# Trust in Intensive Care Patients, Family, and Healthcare Professionals: The Development of a Conceptual Framework Followed by a Case Study

**DOI:** 10.3390/healthcare9020208

**Published:** 2021-02-15

**Authors:** Anne Lotte Lemmers, Peter H. J. van der Voort

**Affiliations:** 1Department of Intensive Care, OLVG Hospital, P.O. Box 95500, 1090HM Amsterdam, The Netherlands; annelottelemmers@gmail.com; 2Department of Critical Care, University Medical Center Groningen, University of Groningen, P.O. Box 30.001, 9700 RB Groningen, The Netherlands

**Keywords:** trust, intensive care, framework, patient, healthcare professional

## Abstract

Intensive care patients experience anxiety, pain, uncertainty, and total dependency. In general, it is important to develop trust between the healthcare professionals (HCPs), patients, and their family. Trust building in the ICU setting is challenging because of the time sensitivity of decision making and the dependency of patients on health care professionals. The objectives of this study are the development of a trust framework and then to use this framework in a case study in the intensive care. In three steps we developed a comprehensive trust framework from the literature concerning trust. First, we identified the elements of trust. Second, we adapted and integrated the dimensions to six concepts to construct the trust framework. Third, these concepts are incorporated into a comprehensive trust framework. In a case study we explored the facilitators and barriers within this framework in eight semi-open interviews with healthcare professionals and eight patients or partners. Trust was first explored inductively and then deductively. We showed that HCPs, patients, and family have largely the same perspective regarding the facilitators of trust, in which communication emerged as the most important one. Other facilitators are maintaining an open feedback culture for HCPs and being aware of patients’ physical and informational privacy. Patients want to be approached as an individual with individual needs. Dishonesty and differences in values and norms were the most important barriers. To contribute to a positive perception of health delivery and to avoid conflicts between HCP and patients or their family we formulated five practical recommendations.

## 1. Introduction

The intensive care is a unique location in the hospital where patients are extremely vulnerable and life-threatening risks are frequently, if not constantly, present. In this setting, patients experience anxiety, fear, pain, and complete dependency on care providers. In addition, family members are emotionally challenged as they fear for the life of their relative and are uncertain about the future.

In this risky and uncertain situation, the development of trust between patients, family and healthcare professionals is important for all of their wellbeing [1,2]. Not only because trust is a precondition for good care but also because mistakes in healthcare can radically affect trust. Distrust in a physician might lead to distrust in the medical system [3], which can increasingly lead to annulments of medical knowledge, a negative patient satisfaction, a malign patient-provider relationship [4], poor medical adherence, not keeping follow-up appointments or following treatment recommendations [5,6,7]. Hence, it is relevant to gain knowledge about trust. Patient-HCP trust is at risk as healthcare systems are growing in a course that deprioritizes one-on-one relationships [8]. This is a result of an increasing demand in care, policy changes and financial pressure [9]. In particular, the acute healthcare system is dealing with an increasing burden which might hinder HCPs to invest in a trust relation with their patients.

Additionally, shifts in welfare, self-consciousness and availability of information have increased people’s knowledge and perceived independence. In line with that HCPs have become advisors instead of decisionmakers in the era of shared decision making (SDM). In SDM, trust plays an important role but trust will only persist when the behavior of HCPs is according to the expectations. As a consequence, HCP should continuously work on maintaining and increasing trust with their patients and families. This is, particularly in the ICU, challenging as it has been shown that ICU professionals score relatively low on emotional empathy, while cognitive empathy is similar to general HCPs [10]. This implies that HCPs working in the ICU understand the situation of patients and family; however, they keep themselves at an emotional distance. This might be explained as a protection measure for their own emotional health [10]. ICU HCPs are constantly exposed to high levels of psychological and physiological distress [11]. How that works out in the development and building of trust between HCPs and patients and their relatives has not been taken into account in a recent review concerning trust in the healthcare setting [12].

## 2. Objectives

The objectives of this study are twofold: first, based on the current literature, we developed a conceptual framework of trust between HCPs and patients with their families which can be applied to the intensive care setting. Second, we explored, in a case study, the facilitators and barriers within this newly developed model in the specific intensive care situation.

## 3. Contextual Background

The healthcare sector is constantly innovating both in techniques and in organization and patient care. The recent change to shared decision making and co-creation is based on the quality of relationships [13,14]. The Dutch Healthcare Consumer Panel has been monitoring patients’ trust in HCPs since 1997. In the Netherlands, the trust of patients in providers is generally very high. However, over the years, there has been noticeable decline in trust [15,16]. Intensive care is part of the acute care chain. A proposed definition of acute care is:


*“Care used to treat sudden, often unexpected, urgent or emergent episodes of injury and illness that can lead to death or disability without rapid intervention.”*
[17,18]

Cases that need acute complex care are more likely to end up on the ICU and are the focus area of this research. Acute care is time sensitive. This implies that HCPs have less or limited time to build a trust relationship. ICU patients are critically ill, often have multiple organ failures and a complex diagnosis. Most patients are sedated and intubated [19,20,21,22], they are completely dependent on the HCPs. The time sensitivity and the dependency of patients to HCP makes it crucial that ICU professionals are able to develop and maintain trust.

## 4. Healthcare Professionals

Healthcare can be provided by various types and levels of employees. Healthcare organization systems often have a hierarchy and seniority is based on clinical experience [23]. This research will make a distinction between nurses and medical specialists (intensivists) and address them in general as HCPs. The ICU nurses and intensivists have followed additional education to be qualified to work on the ICU.

Trust from patient to HCPs is unidirectional; however, it is mutually supportive [24]. In a regulated competition, if HCPs are not trusted by patients, an unwanted consequence might be unwillingness of healthcare organizations and insur to contract them [3,4]. Besides this, HCPs who sense that they are trusted feel more effective.

## 5. Development of a Conceptual Framework

The conceptual framework was constructed in three sequential steps. First, the literature was reviewed concerning the elements of trust. We focused on three main theoretical frameworks by Mayer and Davis, Nooteboom and Hall [25,26,27,28,29]. Mayer and Davis proposed a model of trust in an organizational context [26]. They concluded that the characteristics and actions of trustees determine the level of trust. Nooteboom proposed concepts in the interpersonal context, which shows overlap with the Mayer and Davis framework [28]. In addition, Hall and co-workers have derived the most common dimensions of trust in the context of healthcare and proposed a conceptual model [25]. Table 1 summarizes the dimensions of these three trust frameworks.

Second, we adapted and integrated the dimensions to six concepts to construct the trust framework as is shown in Figure 1.

### 5.1. Competence

The concepts competence and ability (Table 1) are integrated into the framework as they are similarly defined. However, the concept of competence by Hall et al. [25] is limited to communication skills that enhance technical aspects of care, while Nooteboom [28] includes different kinds of competences. Competent medical care involves the collection of medical histories and providing patients with information necessary for a treatment to be effective [25]. As such, for the trust framework competence entails trust in communication skills. Communication is important for the development of trust because it influences the patients’ perspective regarding the HCPs’ competence [25].

### 5.2. Intention

Fidelity and benevolence are incorporated with the concept intention to create a more holistic form of intention. Nooteboom distinguishes between opportunism and lack of dedication or care, both referring to a negative impact on intention [28]. Additionally, Hall et al., differentiates between five indicators of intention [25]. These indicators are characteristics of good intentions. While assessing the intention of HCPs and including either only the negative or the positive side will not provide a complete overview. Therefore, the distinction between opportunism, lack of dedication or care and good intentions will be made.

### 5.3. Integrity

The concepts honesty and confidentiality are included in integrity, which is defined as the degree to which an HCP can be trusted in terms of data, privacy and communicating information [25,28]. Nooteboom distinguished between a lack of honesty and incomplete honesty, while Hall et al., described the characteristics of honesty and the characteristics of confidentiality [25,28].

Honesty is described as ‘the HCP should be telling the truth and avoid lies’. Dishonesty entails the same indicators as lack of honesty or incomplete honesty. They concern lies, half-truths and deception by silence by an HCP. A relevant difference between the two dimensions entails that dishonesty is intentionally and will be denied while incomplete honesty will be acknowledged.

### 5.4. Global Trust

This concept is used only in the theory of Hall et al. [25], as their model is based on the commonality of literature regarding trust in the context of healthcare [25]. This concept will be included in the framework as well. Global trust has two dimensions, the collective concept and holistic aspect of trust. The collective concept addresses issues that have a strong connection with either competence, intention or integrity but they do not completely fit into these concepts. The holistic aspect of trust refers to the expectation that trust has a component which is not dividable [25].

### 5.5. Perceived Risk, Risk-Taking in Relations and Outcome

In Mayer and Davis’ trust theory, the blocks perceived risk, risk-taking in relations and outcome are included [26]. Perceived risk and risk-taking in relations in context of the intensive care setting is the willingness to undergo the risk that the HCPs and treatment might fail. However, HCPs are obligated by law and by their professional oath to provide the best possible care to any patients that arrives. There is a risk of failure and not accepting this risk is, in the critical care setting, usually not an option. Therefore, for the framework, the two blocks will be combined into one block risk.

Trust and risk-taking lead to an outcome. This concept consists of three dimensions [30,31]. Above expectations, which is described as the HCPs exceeded expectations which might lead to an increase in trust. The next dimension, in line with expectation, means that expectations have been met and trust stays at the same level. The last dimension is below expectations, which consists of the option that the expectations were not met by the HCPs and therefore trust can decrease.

In the third step, the concepts are incorporated into a comprehensive trust framework as shown in Figure 2.

## 6. Case Study

### 6.1. Methods

This research is conducted in OLVG-Oost, OVLG has two main locations in Amsterdam, OLVG-East and OLVG-West [20]. OLVG treats more than 500,000 patients per year [21,22]. The ICU in OLVG-East is a mixed medical-surgical 20 bedded unit where 2000 patients per year are treated. The mean age of all ICU patients is 68 years and 60% is male. The patients in this inner-city teaching hospital have a mixed cultural background. In contrast, the nursing and medical staff are predominantly white Dutch healthcare professionals. The intensivists are a mix of anesthesiologists and internists.

Data was collected by means of semi-structured interviews as it allows asking probing questions. They were used to let the respondent expand more in-depth on interesting pathways [32]. At the start of the interviews, respondents were asked to construct a mind map including the factors that they identify as important to trust of patients toward healthcare professionals.

The dimensions of the conceptual framework (Figure 1) were operationalized and summarized in Table 2.

A topic list was constructed based on these dimensions. During the interviews, this topic list was used to guide and assure that every topic was discussed. The interviews were split into two consecutive parts. First, an inductive analysis of essential factors of trust through construction of a mind map was performed. Second, a deductive evaluation of the trust framework’s concepts.

### 6.2. Ethical Considerations

This study was approved by the local committee for medical ethical research (ACWO; OLVG hospital) based on Dutch and European legislation. Participants had to provide informed consent which was obtained according to the guidelines of the ethical committee.

### 6.3. Respondents

The perspective of both ‘patients and family’ and HCPs are studied. One group of interviewees consisted of patients from the ICU. The majority of the patients on the ICU are intubated and, as a result, cannot speak. In those cases, a close relative was interviewed. The other group of respondents consisted of HCPs (physicians and nurses) with different positions in the acute healthcare system. The aim was to interview an equal number of physicians and nurses to identify variations in the perspectives among different healthcare professionals or within their professions. The sampling strategy to select the respondents was purposive sampling. This method allowed to select participants who directly have experience with the research subject [33]. Respondents could be included if they had adequate skills in Dutch or English. The respondents were recruited by email, phone or face-to-face. If the respondents agreed to participate in an interview, they were allowed to choose a place and time to their preference.

### 6.4. Data Analysis

The interview data was analyzed by one of the authors (AL) using content analysis [32]. First, the interviews were carefully transcribed and second, the interviews were coded. Coding was done with the support of Excel^®^ (Microsoft, Redmond, Washington, DC, USA). Deductive codes were derived from the different concepts and dimensions in the trust framework which is presented in the theoretical background. Additionally, inductive coding was used if data did not fit with the pre-determined codes. Two extra themes emerged for the concept integrity. Every transcript was read and coded at least three times to ensure that coding was executed consistently, and no information was missed. Third, to guarantee that the essence of the information was perceived correctly, member checking was included. The summary for member checking also included 2 or 3 quotes on which the respondents could comment if they did not entail the essence of the interview. Finally, the codes were analyzed between stakeholders, within the stakeholder groups and overall.

### 6.5. Results

Sixteen interviews were performed (Table 3). It appeared that all the characteristics described for the trust concepts in the framework, if executed as explained, increase the probability that patients trust or increase their trust in HCPs. As such, these characteristics are facilitators for trust. This immediately suggests that if not executed accurately, they will function as barriers, which was underlined by the respondents. In addition, little differences were found between and within the respondent groups. In the analysis of the results, therefore, no distinctions will be made between groups except if a different perception was present.

The collective mind maps are shown in Figure 3. Patients and partners were clustered in one group. This figure shows that intention and communication were considered to be the most relevant factors for the development of trust in HCPs. None of the shown factors were mentioned by all respondents; however, during the interviews, to the question which competences an HCP should have, all respondents answered with communication as their first or second response.

Comparison of both groups in Figure 3 shows that the ideas of main factors for trust of patients toward HCPs largely overlap between the two groups. Between the groups, the factors were mentioned with the same frequency or with a deviation of one. Three factors that were mentioned by the HCPs were not identified by the patients in the mind maps: human knowledge, acquaintance, and respect. In contrast to the mind maps, they were each mentioned by two patients during the interviews.

Concerning the theoretical framework and its concepts, the factor communication is a sub-category of competence. Coordination would be an element of communication but could also be categorized under the overarching concept competence, because coordination is necessary to achieve the best results. The factors empathy and safety would fit in the concept intention. Furthermore, it speaks volume that HCPs and patients are in line regarding the relevant factors of trust of patient toward HCPs on a basic level. It should be kept in mind that the mind maps were constructed with little explanation from the respondents’ side.

#### 6.5.1. Competence

In line with the results from the mind map, communication was described as most important competence in the context of trust by seven HCPs and six patients. Honesty and respect were mentioned by the remaining respondents. Patients and partners base their trust predominantly on communication skills rather than medical skills and knowledge of HCPs as most patients and family members cannot judge medical expertise. An HCP stated:


*“They (other HCPs ed.) also have to have good communicative qualities. I mean there are doctors who are very smart and can treat very well but they cannot communicate this appropriately to family and even though good actions are done, you will not be able to build trust. (…) a bad doctor can have more trust than a good doctor (Healthcare professional 2).”*


The perspectives of the respondents regarding the characteristics perceived as requisite for communication are shown in Table 4.

Explaining is the most mentioned characteristic of communication competence. The explanation of HCPs should be in line with the level of the patients’ or families’ intelligence, to make sure that they understand. The last attribute confirmatory element concerns asking what patient and family have understood and if they can summarize it. This is useful to assess if the patient and family have understood the HCP correctly and to prevent miscommunication. The next characteristic is personal contact, herein it is important to introduce yourself, explain why you are with a patient, adjusting your energy to the energy of the patient, providing space and acknowledge emotions. These points were addressed by 5 HCP and by six patients. Adjusting energy in terms of speaking volume and emotion is important for personal contact. Therefore, it reveals that it is relevant for HCPs to check their manners and behavior toward patients and family.

From the second dimension, other types of competence, arose knowledge and expertise (mentioned in 14 interviews), knowing and acknowledging personal and knowledge boundaries (7 times), human knowledge (6 times), making time (4 times) and keep promises (4 times).

#### 6.5.2. Intention

Intention was mentioned 11 times in the mind map. Analysis of this concept led to the identification of six characteristics; dedication and effort (mentioned 14 times), attitude and personal attention (13), expectation management (6) and empathy (5). In dedication and effort, comforting refers to HCPs being able to give patients and their family the feeling that they are in good hands and that the HCPs will try to add value to the patients’ life. This was described by HCP1:


*“The most important thing that has to come across is that the patient gets the idea that the physician is going to work for you. You come with a complaint, if it is understood of not. You want to have the feeling that that physician has the goal to find out what is wrong with you and cure that”*


It was emphasized by both HCPs and patients that doing the best for the patient is not always to cure them, it is described as adding value to their lives; however, what value is, is prone to interpretations. Furthermore, patients on the ICU often are sedated and therefore not able to declare their wishes. HCPs indicate that this is challenging, considering that not all families represent the patients’ opinion.

Receiving feedback or criticism and being willing to learn from mistakes is noticed by patients and family as a facilitator for trust. Another characteristic for intention is attitude. HCPs should have an open attitude concerning feedback and approachability. Personal attention (for patients and family) is mentioned as an important feature regarding intention. Herein, it is important to address the patient as an individual.

Non-medical care and the way it is provided should be thorough and with attention. Small gestures and behaviors to patient and their family show a good intention and builds trust.

#### 6.5.3. Integrity

Three themes concerning integrity emerged through inductive coding, which were honesty, privacy, and professionality. They were each mentioned by all respondents. In the mind map, integrity was mentioned 10 times. Honesty is proclaimed by the respondents as one of the most important features for an HCP, moreover, not being honest or lying is perceived as a factor that would greatly decrease trust. Concealing prospects of someone’s disease was judged differently, six respondents from the patient group addressed medical prospects. Four clearly stated they prefer to know every possible scenario and prospect, one explained that being unaware makes it easier to cope with the situation. A patient explained that being lied to about prospects was not appreciated but concealing something is alright as long as it is in the benefit of the patient. All HCPs state that they highly value honesty, one HCP illustrated that in some cultures it is not appreciated being told that their family member is dying or might die. One HCP and one patient pointed out that human knowledge is necessary to assess whether patients or family would like to know all prospects or rather not hear everything.

It is also relevant for trust to admit mistakes. An HCP explained that it is more likely to build credit if you admit a mistake, than concealing it. This was validated and highly valued by two patients, who actually experienced that a mistake was made, in both cases the HCP admitted to the mistake.

Privacy is divided in two subcategories, in terms of information and physical privacy. HCPs are aware of these privacy issues; however they state it is difficult to act accordingly on this. A suggested solution for this issue is private rooms, which would increase both forms privacy. Patient respondents stated that they do not mind this private information issue although they would prefer single rooms. A patient illustrated:


*“Because then, you don’t notice what is happening with other patients, you don’t have to leave for privacy reasons, and the patients are not bothered by other patients”.*


The last characteristic of integrity is professionality. Keeping promises was identified most, both by HCPs and patients. It was emphasized that besides lying, not keeping promises detracts from the integrity and will decrease trust in HCPs. The patient respondents mostly had experiences with nurses not keeping promises, it emerged that they all had to do with poor expectation management. Which suggests that in acute care these are interrelated, since an emergency can always supervene. Concerning small talk and phone use among HCPs, patient respondents specified that it is not wrong, but should be limited. Negative conversations about the job or hospital are not appreciated.

#### 6.5.4. Global Trust

According to two HCPs and three patients respect is a condition for trust. They stressed that mutual respect should be in place before trust can be built.

Therefore, it was emphasized that trust is a feeling and that it is different for every individual, aside from people with mental illnesses which keep them from being able to trust. Additionally, it was explained that physicians often have an ‘automatic’ trust level, while nurses have to put in more effort to build trust.

Regarding the holistic aspect of trust, no significant results came forward from this dimension.

#### 6.5.5. Risk

Respondents identified multiple risks. All previously discussed concepts above could be risks. For instance, good communication helps to build trust, but miscommunication can decrease trust. Three other risks emerged, first, changes in treatment especially those without clear communication about the reason. Patients and family might get the impression that the HCPs are not competent. If it is not clearly explained why the treatment changes are essential. Second, miscommunication and interpretation differences between patients and HCPs but also within families, it was elucidated that this is why it is important to confirm if an explanation is understood. The third risk occurs if there are differences in values and norms of patients, family and their HCP, e.g., cultural differences. The third risk occurs if there are differences in values and norms of patients, family and their HCP. This is illustrated by HCP8:


*“The bulk of the communication disruptions with family, there is a basis of cultural differences why we have a different view regarding what is best for the patient. If it is so fundamentally different, you can understand each other but you will not come together.”*


HCP6 specified that if norms and values of HCPs are not aligned with those of the patient and family, the intention of the HCP clashes per definition. However, it is improbable that HCPs or family will change their values and norms. Interestingly, patients sometimes do change, for example, in case of Muslims or Jehovah witnesses who when they are almost dying and in a lot of pain do want morphine or blood. Hence, differences in norms and values is a complex risk to solve for trust.

#### 6.5.6. Outcome

Outcome was analyzed by means of the three dimensions (1) above expectations, (2) in line with expectations and (3) below expectation. Except for one patient, the respondents stated that they did have trust. Moreover, five of eight patient/family respondents stated that their trust had increased during the stay on the ICU and seven of the patient respondents emphasized that they were satisfied with the medical care. Even though some respondents experienced mistakes made by HCPs they explained that they are also are human and are allowed to make a mistake. This illustrates that if trust is in place, a mistake, especially if the HCPs are honest about it, is not an immediate reason for a decrease of trust

However, trust issues do arise on the ICU, according to two HCPs this is influenced by the outcome, if the patient leaves healthy or dies. Six HCPs explained that patients and their families who have a prolonged length of stay in the ICU, have complex illnesses and find it relatively frequently difficult to maintain trust. They explained that this might be related to one or more of the three risks as explained above.

## 7. Discussion

We studied the facilitators and barriers to build trust in acute healthcare, especially the intensive care unit, by first defining a conceptual framework based on the current literature and second exploring this model in a qualitative field study. Facilitators and barriers were found in the dimensions competence, intention, integrity, global trust, risk and outcome. Learning from mistakes and being open to feedback appears to be a facilitator for HCPs. This result is in line with earlier research in which it was found that the course of action of HCPs regarding mistakes influences how patients feel after an adverse event or complication. Moreover, patients were less upset if the HCP apologized and revealed the mistake through honesty and with compassion [34]. Oates underlines the relevance of a culture where honesty and openness exist [35]. This relates to admitting mistakes and errors that are openly reported and treated as opportunities for organizational learning and improvement [35]. This kind of a feedback-friendly culture builds trust [36,37]. Thus, to increase trust, a culture of feedback helps. Ramani et al., identified three themes regarding a change of feedback culture [38]. First, normalizing constructive feedback to promote a culture of growth, second, overcoming the mental block to feedback seeking, and third, a hierarchical culture impeding bidirectional feedback.

Another facilitator that we identified is paying attention to the patient as an individual, mainly personal attention. This refers to the approach that HCPs should have toward the patient and family. The patient is not only a patient, it is also a human being and should be approached as one [39]. Moreover, recent research ‘In search of mangomoments’, stated that mangomoments, which are small, unexpected acts or gestures, with great value for the experience of care of patients, families and HCPs, bring happiness to the patients and families, it might even decrease burnout risk of HCPs because of an increased joy in their work [40].

Brown et al., determined that ICU patients are susceptible to dehumanization because they lack the characteristics of human beings; consciousness, agency and self-determination [41]. These authors also revealed that dehumanization can occur without clinicians’ awareness, due to the complex and demanding setting on the ICU. Nevertheless, this issue is growing in relevance because healthcare is transitioning to deprioritizing on one-on-one relationships and personal attention [8,42]. Which aligns with the finding that a lack of personal attention decreases patients experience of HCPs’ intention and therefore might decrease trust.

From the integrity concept arose that private information should be handled discretely. Earlier research stated that privacy during hospitalization can be endangered because the setting is often intimate [43]. Ozturk [44] explained that privacy protection is closely related to patients’ trust in HCPs [44]. Additionally, Rojas showed that HCPs usually defend the privacy in terms of the identity of the patients, while they neglect the feelings and thoughts a patient might have [45]. Nevertheless, this does not incline that the privacy in terms of identity or personal information is always preserved.

The ‘ICU of the future’ consists, according to Halpern and Vincent of single rooms, due to the enhanced opportunities for privacy protection and infection prevention [46,47]. Watson and co-workers studied the morning transfer, which is susceptible for privacy issues, and compared beside rounds with table rounds on the ICU [48]. It was concluded that table rounds were correlated with a perception of increased privacy and an increased overall satisfaction of multidisciplinary team members [48]. These findings directly connect with our findings. Patient privacy and trust building contradicts with the HCPs’ statements that they prefer to be able to see the patients directly at all times.

The most frequently mentioned barrier to build trust by both patients and HCPs was dishonesty. This insight is in line with Shoemaker who determined that in a trustful patient-HCP relationship, the patient should assume that they will be treated with respect and honesty [34]. Moreover, trust can turn into mistrust, when someone carries out a deed of treason or failed to maintain their promises [49]. As mentioned before, limited ability in interpersonal skills can strengthen mistrust. Trust is therefore often related to providing information honestly [50]. Trust is also associated with keeping promises [51,52]. This indicates that in order to maintain patients’ trust, HCPs should be honest in terms of communication, e.g., about prospects, promises and regarding mistakes.

The second barrier that was identified is differences in values and norms of patients. This can be an issue, for instance, when HCPs decide to initiate treatment limitations such as the switch to palliative care. Every individual has its own cultural, ethical and religious context [53]. This may lead to divergent values and norms, which can cause ethical dilemmas and indicate multiple courses of action [54]. In severe ethical dilemmas it is unavoidable that at least one value will be lost [54]. Henderson and co-workers proposed a framework for cultural competence, which entails the understanding of HCPs to respect and tailor healthcare in a diverse cultural encounter [55]. They concluded that cultural competence results in improved health outcomes, perceived quality healthcare, satisfaction with healthcare and treatment adherence [55]. In addition, on the ICU, cultural competency training might contribute in providing high quality care [56].

From the concept global trust, no facilitators or barriers emerged. This concept merely provided preconditions and contextual understandings. Both respondent groups mentioned mutual respect. Not only for the patient but also for the healthcare professional. Currently, respect for HCPs is decreasing which might negatively influence trust as well [57]. Respect for HCPs used to come from authority. However, respect presently emerges from the knowledge and personality of HCPs [57].

This study has both strengths and limitations. A strength of this research is the mixed methods, the inductive and deductive approach lead to methodological triangulation and contributed to the validity of this research [32]. Although the semi-structured interviews were appropriate, it is difficult to assure that no biases arise [32]. Moreover, relatively few patients were included compared to HCPs and family members due to availability and severity of disease. Patients in the ICU may suffer from delirium which could contribute to recall bias [58]. Interviewing family members bypassed that problem.

The monocenter design may not ensure extrapolation of the results to other ICUs; however, this ICU is a mixed medical and surgical ICU comparable with most ICUs in the western world. The inclusion of HCPs, patients and family created the unique opportunity to detect discrepancies in the perception. Although we included a relatively small number of subjects, we randomly choose them from the ICU population, both patients and HCP. In addition, the interviews showed saturation. Both are arguments that the research is representative for the population.

This research shows that knowledge of facilitators and barriers for trust can be important in daily care. We provide a framework for both practical use and future research in the intensive care setting. Trust can probably contribute to a high level of satisfaction for patients and families and also for the health care professionals. It probably helps in avoiding conflicts with patients and their family. Conflicts are a known key risk factor for burnouts among HCPs [59,60], which is associated with a decreased quality of care, lower patient satisfaction and high rates of staff turnover [61]. Complying to the facilitators that we identified and using them in practice, will increase trust between patients, family and their healthcare providers in the complex intensive care setting.

## 8. Conclusions

In this study, we built a trust framework from available literature and defined the facilitators and barriers of trust from the perspectives of patients, families, and HCPs in the intensive care setting. From a case study based on this framework, we conclude that (1) patients want to be treated as individuals (2) both physical and informational privacy of patients should be protected (3) communication is critical (4) an open atmosphere where mistakes can be admitted blame free and are used to learn from, appears to be beneficial for the trust patients have in HCPs and (5) trust issues mostly arise from miscommunication or from having different values and norms (culture) between HCP and patients or their family.

## 9. Recommendations

### 9.1. Short-Term

#### 9.1.1. Change the Feedback Culture

Almost all HCP respondents addressed that it is currently difficult to give feedback because it is not the accepted culture on the ICU. Constructing a plan to change the current feedback culture into a feedback-friendly culture, in which feedback can be provided bottom-up and top-down. This will be helpful for both patients and HCPs, as the first notice that the atmosphere is easy, and HCP do everything to become the best version of the HCP they could be.

#### 9.1.2. Humanize Intensive Care

Partners of patients explained that patients can grow extremely happy after a small moment of personal attention and feeling like a person instead of a patient. Therefore, additional attention should be given to patients to increase their experience and their trust.

#### 9.1.3. Intensify the Education on Communication of HCPs

Communication is addressed as the most important competence for an HCP to acquire. Patients base trust on communication skills more than on medical knowledge and medical skills. Hence, due to the relevance of this competence it is recommended to constantly keep improving communication skills, even after finishing the primary education.

### 9.2. Long-Term

#### 9.2.1. Single Person Rooms

Both patients and HCPs addressed that two-person rooms can decrease privacy and especially the silence on the ICU. To improve privacy hazards and increase the rest patients get on the ICU, it is recommended that in the future rooms are designed to be occupied by one patient.

#### 9.2.2. Create Opportunities for Patients to Get More Rest

Machines in the ICU make a lot of noise, not only in the normal state but they also have alarms going off if changes happen. Therefore, patients are prone to too little rest, which could negatively influence the recovery. Hence, it will be relevant to research how the machines on the ICU can be made being more silent.

## Figures and Tables

**Figure 1 healthcare-09-00208-f001:**
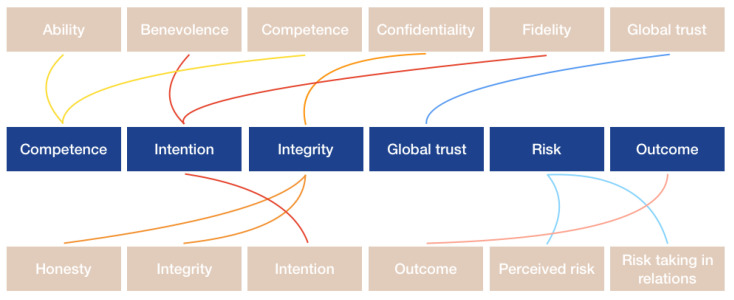
Integration of the concepts into the trust framework. On the upper and downside of the figure the concepts of trust according to Mayer & Davis [26], Nooteboom [28] and Hall et al. [25] are displayed, as mentioned in Table 1. In the middle row, the concepts are integrated into the concepts of the proposed trust framework.

**Figure 2 healthcare-09-00208-f002:**
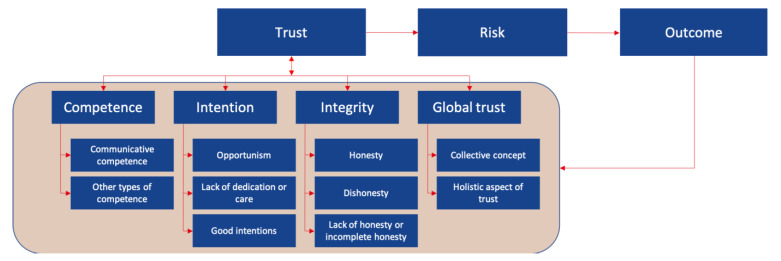
Trust conceptual framework including the six integrated concepts with their dimensions.

**Figure 3 healthcare-09-00208-f003:**
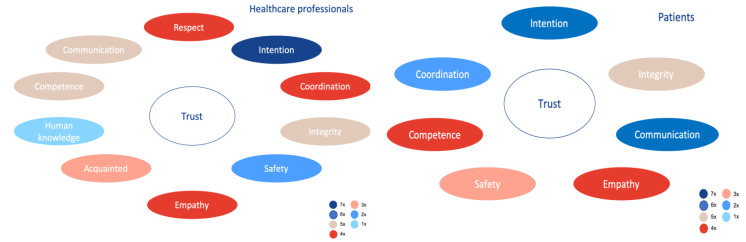
An overview of the main factors for trust mentioned by healthcare professionals. The colors in the figure relate to how often a factor was addressed. 1× means: once mentioned in the interviews, 2×: twice mentioned, etcetera.

**Table 1 healthcare-09-00208-t001:** Overview of the concepts of Mayer and Davis [26], Hall et al. [25] and Nooteboom [28,29].

	Mayer & Davis [26]	Hall et al. [25]	Nooteboom [28,29]
Ability	The group of skills, competencies, and characteristics that enable a party to have influence within some specific domain	-	-
Benevolence	The extent to which a trustee is believed to want to do good to the trustor, aside from an egocentric profit motive	-	-
Competence	-	Producing the best achievable results and avoiding mistakes	Including but not exclusively, skills and knowledge concerning production of goods or services, employing technology or building and maintaining relations with other people
Confidentiality	-	Proper use of data and privacy	-
Fidelity	-	Acting in the best interest of the patient and not taking advantage of their vulnerability	-
Global trust	-	- A collective concept for concerns which have a strong connection with some areas but do not fit in one	-
		- A more holistic aspect of trust, as it is reasonable to expect that trust has a significant component which is not divisible	
Honesty	-	Telling the truth and evasion of intentional lies	Trust in the truthfulness of an actor, this actor often is the only information source
Integrity	The relationship between integrity and trust involves the trustor’s perception that the trustee adheres to a set of principles that the trustor finds acceptable	-	-
Intention	-	-	The expectation that the partner will not behave opportunistically
Outcome	The outcome of the trusting behavior (favorable or unfavorable) will influence trust indirectly through the perceptions of ability, benevolence, and integrity at the next interaction	-	-
Perceived risk	Involves the risks that are identified by trustee and trustor	-	-
Risk-taking in relations	differentiates the outcomes of trust from general risk-taking behaviors because it can occur only in the context of a specific, identifiable relationship with another party	-	-

**Table 2 healthcare-09-00208-t002:** Operationalization table; with the six integrated concepts, their dimensions, and operational definitions of the trust conceptual framework.

Concept	Dimensions	Operational Definition
Competence	Communication competence	The perception of the respondent regarding the ability of the HCP build and maintain relations with other people as well as being able to make themselves understood, and understand others.
	Other types of competence	The perception of the respondent regarding skills and knowledge concerning production of goods or services, employing technology Intention.
Intention	Opportunism	The perception of respondents regarding endangerment of the relationship by the HCP.
	Lack of dedication or care	The perception of the respondent regarding the effort and attention shown by the HCP.
	Good intentions	The perception of the respondent regarding the loyalty, caring, respect, advocacy or avoiding conflicts of interest by HCP.
Integrity	Honesty	The perception of the respondent regarding if the HCP tells the truth and avoids lies.
	Dishonesty	The perception of the respondent regarding if HCP lie, or tell half-truths, even if they are confronted with their lies/deceptions.
	Lack of honesty or incomplete honesty	The perception of the respondent regarding if HCP lie or tell half-truths.
Global trust	Collective concept	The perception of the respondent regarding concerns which have a strong connection with some areas but do not fit in one.
	Holistic aspect of trust	The perception of the respondent regarding the expectation that trust has a significant component which is not divisible.
Risk		The perception of the respondent regarding the risks they have undergone in the trust relationship.
Outcome	Above expectations	The perception of the respondent regarding the exceeding of the expectations.
	In line with expectations	The perception of the respondent regarding if the expectations have been met.
	Below expectations	The perception of the respondent regarding the failing of expectations.

**Table 3 healthcare-09-00208-t003:** Respondents’ characteristics (Length of Stay (LOS)).

Respondent	Patient/Partner	Male/Female (Patient)	Age (Patient)	LOS (on the ICU)/Time of Employance
	Intensivist/Nurse			
Patient 1	Patient + partner	F	38	28 days
Patient 2	Patient	M	67	8 days
Patient 3	Partner	M	64	6 days
Patient 4	Partner	F	30	1 day
Patient 5	Partner	M	51	10 days
Patient 6	Partner	M	62	9 weeks
Patient 7	Partner	M	37	5 weeks
Patient 8	Patient	F	68	2 days
HCP 1	Nurse	F	-	-
HCP 2	Nurse	M	-	31 years
HCP 3	Intensivist	M	-	2 years
HCP 4	Intensivist	F	-	16 years
HCP 5	Nurse	F	-	15 years
HCP 6	Intensivist	M	-	2.5 years
HCP 7	Intensivist	M	-	10 years
HCP 8	Intensivist	F	-	14 years

**Table 4 healthcare-09-00208-t004:** Respondents’ perspectives regarding the characteristics of proper communication. P/F; patient or family. HCP; health care professional.

Communication Competence	Attributes	Total (P/F; HCP)
Explaining	Should be clear, fitting to the patients/partners preferences and intelligence, and have a confirmatory element	15 (7:8)
Personal contact	The behavior and courtesy of HCPs	11 (6:5)
Medical contact/information	The clarity regarding where or with whom patients/family should receive medical information	8 (6:2)
Coordination	Alignments between HCPs regarding patient communication	6 (5:1)
Listening	Hear what the patient and/or family explain	7 (3:4)
Relevance of communication	Understanding of the power of communication	8 (2:6)

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
