# Peer review of "Trust in Intensive Care Patients, Family, and Healthcare Professionals: The Development of a Conceptual Framework Followed by a Case Study"

_healthcare, 2021, doi:10.3390/healthcare9020208_

Round 1

Reviewer 1 Report

Thank you for exploring this important of trust in care within the ICU. 

This paper makes an interesting contribution to the field, and adds to learning about the relationship between health care professionals and the people within their care. I have made some suggestions that would strengthen the paper. 

Title: The title could be changed to clarify the paper, e.g. development of framework and case study.

Introduction:

Please introduce the context of the research, the location, community it serves, number of beds, staff mix etc. 

Background:

Why this was needed? What are the local issues, given it is a case study. 

Demographics of the population being studied. Age, gender, cultural background, type of illness, specialty of intensivists, types of training. 

Methodology:

The results suggest that this research falls under the case study methodology as it explores these ideas within a single ICU with particular characteristics.

The authors could define this context up front in the paper, and within the methodology section. 

This paper would be improved with the consistent use of terminology to define participants within he setting, sometimes words such as agent and actor are introduced without defining what this means, it seems out of context for this paper.

Conclusions 

Conclusions could be strengthened once the context is more clearly defined and explored.

Overall

The paper provide a general overview, the reader requires more in-depth information about the research specifically, to be able make decisions about whether the results can be applied in different contexts. This is an important issue which could be addressed globally however this is unclear due to the lack of detail within he paper. 

Author Response

Reviewer 1

Title: The title could be changed to clarify the paper, e.g. development of framework and case study.

Answer: we have changed the title accordingly and used the word case study elsewhere in the manuscript as well.

Introduction: 

Please introduce the context of the research, the location, community it serves, number of beds, staff mix etc. 

Answer: we have added data concerning the hospital, the community, the patients and the nursing and medical staff in the section ‘contextual background’. We also added a paragraph underlining the importance of trust between patient and HCP. Illustrating the possible effects on the healthcare system if trust would grow to distrust.

Background:

Why this was needed? What are the local issues, given it is a case study. 

Demographics of the population being studied. Age, gender, cultural background, type of illness, specialty of intensivists, types of training. 

Answer: we have now added local information and information about the population as requested. A contextual background paragraph is added to give insight in the characteristics of the ICU and the patients treated; to address a short history of patient-HCP relationship including the fact that currently trust in HCPs in hospitals is high; to explain about the acute care system; to address the context of the research, the hospital in which the research was conducted, and which HCPs are included and to explain that the trust relation is not only important for the patients and their family but also for the HCPs.

Methodology:

The results suggest that this research falls under the case study methodology as it explores these ideas within a single ICU with particular characteristics.

The authors could define this context up front in the paper, and within the methodology section. 

Answer: we have clarified this issue now in a section ‘objectives’

This paper would be improved with the consistent use of terminology to define participants within he setting, sometimes words such as agent and actor are introduced without defining what this means, it seems out of context for this paper.

Answer: we have read and adapted the paper to improve the consistency in terminology.

Conclusions 

Conclusions could be strengthened once the context is more clearly defined and explored.

Answer: we have now rephrased and strengthened the conclusion and included recommendations with short and long term suggestions to improve trust.

Overall

The paper provide a general overview, the reader requires more in-depth information about the research specifically, to be able make decisions about whether the results can be applied in different contexts. This is an important issue which could be addressed globally however this is unclear due to the lack of detail within he paper. 

Answer: in the paper we have now on several locations provided more detailed information. We have added information about the local situation, the ICU population in general, the HCP in the ICU, a table with characteristics from the patients and HCP and we have added some quotes from the interviews. In addition, we have added direct and practical recommendations.

Reviewer 2 Report

Firstly, to congratulate them for their contribution and to thank them for making it available to the scientific community.
I have been very interested in your work and therefore I dare to make the following formal suggestions in case you would like to attend to them. I have divided them into three sections: research objectives, methodology and results.
Objectives of the research
In my opinion, the objectives could be more clearly defined in the introduction section, including the questions that have guided the study.
I understand that the research has a twofold objective: (1) to develop a conceptual framework of trust between health professionals, patients, and their families in the context of intensive care oriented health services, based on previous input on the topic, and (2) according to this model, to explore barriers and facilitators for the development of trust through some interviews with patients and health professionals.
In relation to the first of the objectives, I would like to make the following observation: Undoubtedly, any person sensitive to these issues knows the importance of mutual trust between health care providers and recipients for a better provision of these services and, consequently, a better recovery of the patient's health, but apart from this pre-existing general knowledge, I understand that they should make their objective more explicit in the introduction and, above all, justify its practical relevance by providing evidence in this regard.
Secondly, given that they use as a basis for their conceptual proposal theoretical models that have been formulated in different fields of knowledge, they should explicitly communicate their arguments to make the transposition of concepts from one field to another, since it is not clear to me how to move from an organizational context to an interpersonal one.
Methodology used
The methods used are adequate to achieve the established objectives. However, although interviews and content analysis are adequate techniques to investigate the factors that, according to the persons involved (professionals, patients, and their families), hinder or facilitate the development of trust, I believe it would be necessary to provide information about the validation of the coding process and the degree of agreement among judges when coding the relevant information from the interviews. In addition, I suggest that the categories used in the coding of the information be exemplified with textual quotations from the interviewees' responses.
On the other hand, given that the number of people interviewed is small, I think it would be good if they explicitly stated why they believe that the participants in their study are representative cases of those we can find in the provision and perception of intensive care, since this would give greater support to their conceptual proposal, insofar as it affects the extensibility or generalization of their conclusions.
Applicability of the results
I think I have understood well the purpose of your proposal, but this one, in my opinion, would be more interesting if it were accompanied by a guide or guidelines for its practical application in an intensive care unit.

Author Response

Reviewer 2

Objectives of the research
In my opinion, the objectives could be more clearly defined in the introduction section, including the questions that have guided the study.
I understand that the research has a twofold objective: (1) to develop a conceptual framework of trust between health professionals, patients, and their families in the context of intensive care oriented health services, based on previous input on the topic, and (2) according to this model, to explore barriers and facilitators for the development of trust through some interviews with patients and health professionals.

Answer: we have now provided a short section ‘Objectives’ to clarify this issue.

In relation to the first of the objectives, I would like to make the following observation: Undoubtedly, any person sensitive to these issues knows the importance of mutual trust between health care providers and recipients for a better provision of these services and, consequently, a better recovery of the patient's health, but apart from this pre-existing general knowledge, I understand that they should make their objective more explicit in the introduction and, above all, justify its practical relevance by providing evidence in this regard.

Answer: we have now added paragraphs to introduction and contextual background underlining the importance of trust between patient and HCP. This illustrates the possible effects on the healthcare system if trust would grow to distrust and it makes the connection between system, organization and patient/family.

Secondly, given that they use as a basis for their conceptual proposal theoretical models that have been formulated in different fields of knowledge, they should explicitly communicate their arguments to make the transposition of concepts from one field to another, since it is not clear to me how to move from an organizational context to an interpersonal one.

Answer: In the introduction and in a contextual background paragraph we have now added text to address this issue. We describe the acute care organization with its unique time-sensitivity and patient-HCP dependency. In that role, HCP should be able to build and maintain a high level of trust effectively in a short period of time.

Methodology used
The methods used are adequate to achieve the established objectives. However, although interviews and content analysis are adequate techniques to investigate the factors that, according to the persons involved (professionals, patients, and their families), hinder or facilitate the development of trust, I believe it would be necessary to provide information about the validation of the coding process and the degree of agreement among judges when coding the relevant information from the interviews. In addition, I suggest that the categories used in the coding of the information be exemplified with textual quotations from the interviewees' responses.

Answer: the method section is completed on several points. Both deductive coding and inductive coding is mentioned. Inductive coding resulted in two extra themes for the concept integrity.

On the other hand, given that the number of people interviewed is small, I think it would be good if they explicitly stated why they believe that the participants in their study are representative cases of those we can find in the provision and perception of intensive care, since this would give greater support to their conceptual proposal, insofar as it affects the extensibility or generalization of their conclusions.

Answer: we have addressed this issue now in the discussion section under ‘limitations’.

Applicability of the results
I think I have understood well the purpose of your proposal, but this one, in my opinion, would be more interesting if it were accompanied by a guide or guidelines for its practical application in an intensive care unit.

Answer: we have now added 5 practical recommendations with short and long term suggestions to improve trust.

Round 2

Reviewer 2 Report

I would like to thank you for this rewriting of your work, which answers all the concerns that I had when I first read the manuscript. In all sincerity, I believe that this rewriting clearly states the objectives of the study and its rationale. It resolves the methodological doubts raised and gives the work a practical character that it lacked before. I therefore thank you for your interest in resolving the doubts raised and congratulate you on the work carried out.